# *Vibrio* Species in an Urban Tropical Estuary: Antimicrobial Susceptibility, Interaction with Environmental Parameters, and Possible Public Health Outcomes

**DOI:** 10.3390/microorganisms9051007

**Published:** 2021-05-07

**Authors:** Anna L. B. Canellas, Isabelle R. Lopes, Marianne P. Mello, Rodolfo Paranhos, Bruno F. R. de Oliveira, Marinella S. Laport

**Affiliations:** 1Laboratório de Bacteriologia Molecular e Marinha, Instituto de Microbiologia Paulo de Góes, Universidade Federal do Rio de Janeiro, Rio de Janeiro 21941902, Brazil; annaluiza@micro.ufrj.br (A.L.B.C.); rodrigueslopes.isabelle@gmail.com (I.R.L.); bfroliveira@micro.ufrj.br (B.F.R.d.O.); 2Departamento de Biologia Marinha, Instituto de Biologia, Universidade Federal do Rio de Janeiro, Rio de Janeiro 21941617, Brazil; maripmello17@gmail.com (M.P.M.); ufrj.rodolfo@gmail.com (R.P.)

**Keywords:** antimicrobial resistance, Guanabara Bay, marine pollution, public health, *Vibrio*

## Abstract

The genus *Vibrio* comprises pathogens ubiquitous to marine environments. This study evaluated the cultivable *Vibrio* community in the Guanabara Bay (GB), a recreational, yet heavily polluted estuary in Rio de Janeiro, Brazil. Over one year, 66 water samples from three locations along a pollution gradient were investigated. Isolates were identified by MALDI-TOF mass spectrometry, revealing 20 *Vibrio* species, including several potential pathogens. Antimicrobial susceptibility testing confirmed resistance to aminoglycosides, beta-lactams (including carbapenems and third-generation cephalosporins), fluoroquinolones, sulfonamides, and tetracyclines. Four strains were producers of extended-spectrum beta-lactamases (ESBL), all of which carried beta-lactam and heavy metal resistance genes. The *toxR* gene was detected in all *V. parahaemolyticus* strains, although none carried the *tdh* or *trh* genes. Higher bacterial isolation rates occurred in months marked by higher water temperatures, lower salinities, and lower phosphorus and nitrogen concentrations. The presence of non-susceptible *Vibrio* spp. was related to indicators of eutrophication and sewage inflow. DNA fingerprinting analyses revealed that *V. harveyi* and *V. parahaemolyticus* strains non-susceptible to antimicrobials might persist in these waters throughout the year. Our findings indicate the presence of antimicrobial-resistant and potentially pathogenic *Vibrio* spp. in a recreational environment, raising concerns about the possible risks of human exposure to these waters.

## 1. Introduction

The genus *Vibrio* includes Gram-negative curved rod-shaped Gammaproteobacteria autochthonous in aquatic environments, where they play important roles in nutrient cycling [1]. These bacteria comprise a remarkably diverse group and are known not only for their ability to rapidly adapt to environmental changes but also for their intrinsic competency to acquire and transmit exogenous DNA via horizontal gene transfer (HGT) events, especially among pathogenic strains [2]. *Vibrio* spp. can be found either in a free-living state or associated with marine animals and particles, such as plastics and phytoplankton [1,3]. Currently, there are 123 validated species according to the Genome Taxonomy Database [4]. From these, *Vibrio cholerae*, *Vibrio vulnificus*, *Vibrio parahaemolyticus*, and *Vibrio alginolyticus* are the four species, also known as the “big four”, often associated with human infections [5].

*Vibrio* spp. infections usually occur via the consumption of contaminated seafood or exposure to contaminated water and tend to be self-limiting. However, they can be fatal, especially in immunocompromised patients or upon failure of antimicrobial therapy [6,7]. Although cholera cases are mostly reported in areas where water quality is inadequate and sewage treatment insufficient, other *Vibrio* species are relevant agents of seafood-borne infections on a global scale. In fact, the genus *Vibrio* along with 13 other pathogens are accountable for more than 95% of foodborne infections and subsequent hospitalizations in the United States and responsible for 98% of deaths [8]. Moreover, extraintestinal infections caused by *Vibrio* spp., such as ear and wound infections, have been gaining more attention in view of the consequences of climate change, which have been linked to a rise in such cases [5,9]. There is currently an absence of surveillance systems dedicated to *Vibrio* illnesses outside the United States, a country where these data have been gathered since 1989 through “The Cholera and Other *Vibrio* Illness Surveillance” (COVIS) system [10]. Hence, there may be an underestimation of *Vibrio* spp. infections worldwide. Albeit usually susceptible to most of the currently used antibacterial drugs, resistant *Vibrio* spp. have been reported with increasing frequency in many countries over the past years, raising concerns about the remaining treatment options and food safety [11,12].

Although widely investigated under the clinical framework, there is a significant knowledge gap regarding the antimicrobial resistance issue in environmental settings, notably aquatic ecosystems. The presence of clinically relevant traits in environmental bacteria, such as certain antimicrobial and heavy metal resistance genes, poses a challenge to public health and supports the hypothesis that anthropogenic action is one of the most important evolutionary forces on the planet [13]. Since aquatic environments often serve as recreational areas and sources of subsistence to the population, it is fundamental to better understand the risks they may pose to human health, in particular their active role as hotspots of virulent and/or antimicrobial-resistant microorganisms [14].

Potentially pathogenic *Vibrio* spp. have been previously isolated in the Guanabara Bay (GB), Rio de Janeiro, Brazil [15,16]. However, little attention has been paid to the investigation of their antimicrobial susceptibility and monitoring of their spatio-temporal dynamics in this ecosystem. The GB is a eutrophic estuarine system located in a humid tropical region surrounded by the second major metropolitan area in Brazil. GB is the second-largest bay on the Brazilian coast with a surface area of approximately 384 km^2^ and a volume of 1.87 × 10^9^ m^3^ [17]. In its surroundings, there are more than 15 cities, thousands of industries, refineries, two ports, and several shipyards. In addition, circa 10 million people live in GB’s surroundings, from which around 4 million live in informal settlements with precarious sanitary and urbanization conditions. On account of the lack of proper sewage collection and treatment, the GB is chronically impacted by untreated sewage (~18 m^3^/s), petroleum residues, littering, and the discharge of industrial, rural, and hospital effluents [18,19]. Despite the fact that the GB receives water from the Atlantic Ocean and several rivers and streams, its inner regions are often considered inappropriate for bathing according to local monitoring services [20]. Nonetheless, navigation, fishing, and swimming are common activities in the area.

The GB shares some characteristics with other impacted environments in the world, such as the Bay of Bengal, known to be vulnerable to water-borne diseases and pollution [21]. In 2018, local monitoring systems in Rio de Janeiro’s metropolitan area have reported more than 1200 hospitalizations and more than 30 deaths on account of water-borne illnesses [22]. Also, many studies worldwide indicate that individuals who are directly in contact with recreational waters, e.g., swimming or fishing, are more likely to develop certain illnesses, such as gastrointestinal infections [23,24]. Despite being a major aquatic ecosystem in this region, little is known about the risks of exposure to GB’s waters to human health. Therefore, this study aimed to investigate the cultivable *Vibrio* community in three distinct locations along a pollution gradient in the GB over a one-year period (2018–2019). By doing so, we observed that potentially pathogenic *Vibrio* strains are present in GB’s waters and that they can be resistant to multiple antimicrobials. DNA fingerprinting analyses further indicated that groups of resistant strains belonging to the same species are present in the GB throughout the year. Moreover, the most ecologically impacted location presented not only the highest bacterial abundance but also the highest rates of antimicrobial-resistant bacteria harboring beta-lactam and heavy metal resistance genes. Correlation analyses suggest that *Vibrio* abundance is linked with eutrophication and sewage inflow indicators. The occurrence of virulence markers was also evaluated, raising important questions concerning the current water monitoring protocols and risk-assessment research to estimate the possible public health outcomes. Up to now, we believe this is the first study dedicated to the description of *Vibrio* species and their antimicrobial resistance traits in the GB.

## 2. Materials and Methods

### 2.1. Water Sampling and Processing

Subsurface (1–2 m depth) and bottom waters (6–20 m depth) were sampled monthly from three sites in the GB, Rio de Janeiro, Brazil, from June 2018 to May 2019, totalizing 66 water samples (33 from the subsurface and 33 from the bottom) (Appendix A). Each sampling site differed in terms of its proximity to the ocean and degrees of anthropogenic impact. The first site (1: 22°55′43″ S; 43°08′51″ W) is located near the entrance of the Bay and is under strong influence from the Atlantic Ocean. The second site (7: 22°52′12″ S; 43°09′46″ W) is subjected to the mixing of inner and oceanic waters. Therefore, it is considered the intermediate site. Finally, the third site (34: 22°50′09″ S; 43°14′56″ W) is the closest to land and is known to receive significant effluent discharge. We also highlight the existence of sewage channels, especially in the proximity of site 34, which contribute to the discharge of sewage *in natura* [15].

All sampling expeditions took place in the morning period and on days with little or no tidal influence. At each sampling site, water samples from the subsurface and the bottom were collected using a 5 L Niskin bottle (General Oceanics, Miami, FL, USA), stored in appropriate sterile plastic bottles and immediately transported to the laboratory in an icebox. Samples from each sampling site and expedition were analyzed and the following parameters were determined: water and air temperature, water transparency, salinity, total phosphorus, total nitrogen, chlorophyll, pheophytin, thermotolerant and total coliforms, as well as *Escherichia coli* and heterotrophic bacteria counts. Water samples were assessed for levels of microbiological, physical, and chemical parameters by using previously described and well-established oceanographic methods [25,26,27]. Water temperature and salinity were measured using a CTD device (Sea-Bird Electronics, Inc., Bellevue, WA, USA). In two sampling expeditions (June and October 2018), the water temperature could not be measured due to CTD malfunction. Transparency was determined with a Secchi disc. Concentrations of total phosphorus were determined by acid digestion to phosphate and concentrations of total nitrogen were determined by digestion with potassium persulfate following nitrate determination [15]. 

### 2.2. Vibrio Isolation

Monthly water samples (100 µL) from each sampling site (subsurface and bottom) were directly plated onto Thiosulfate Citrate Bile Salts Sucrose agar (TCBS, Isofar, Rio de Janeiro, RJ, Brazil). Colonies (yellow, green, or blue smooth round-shaped colonies) were selected after overnight incubation at 25 °C and then inoculated onto Marine agar (Difco, Franklin Lakes, NJ, USA) supplemented with ampicillin (16 µg/mL). Since high resistance rates to penicillins have been repeatedly reported in *Vibrio* spp., ampicillin was employed in order to enhance the selection of this bacterial genus [28]. All grown colonies were finally stored in Marine broth with 30% glycerol (*v/v*) at −20 °C for further analyses.

### 2.3. Identification and Characterization of Vibrio

The viable isolates were identified based on the Gram staining method and catalase test [29]. Isolates characterized as Gram-negative bacilli and catalase-positive were submitted to identification by Matrix-Assisted Laser Desorption/Ionization Time-of-Flight Mass Spectrometry (MALDI-TOF MS) on the Microflex LT MS platform (Bruker Daltonics, Bremen, Germany). Samples were prepared as previously described [30]. The obtained mass spectra were compared to the references in the database using MALDI Biotyper 7.0 program (Bruker^®^, 2019). The score values obtained were interpreted according to the manufacturer’s guidelines: ≥2.300 indicated reliable identification at species level; 2.000–2.299 reliable identification at genus level and likely identification for species; 1.700–1.999 reliable identification solely at genus level; and <1.699 was not considered reliable for identification. *E. coli* DH5α was used as a quality control strain.

### 2.4. Antimicrobial Susceptibility Testing

Antimicrobial susceptibility of viable *Vibrio* spp. isolates was determined using the disk diffusion method on Mueller-Hinton agar plates (Kasvi, São José dos Pinhais, SP, Brazil) according to the guidelines of the Clinical and Laboratory Standards Institute (CLSI, 2010) since it is the last edition containing well-defined breakpoints for the genus *Vibrio* [31]. In this study, eight antimicrobials (Sensidisc, São Paulo, SP, Brazil) belonging to five different classes (aminoglycosides, beta-lactams, fluoroquinolones, sulfonamides, and tetracyclines) were tested: amikacin (AMI), amoxicillin-clavulanic acid (AMC), cefotaxime (CTX), ceftazidime (CAZ), ciprofloxacin (CIP), imipenem (IPM), trimethoprim-sulfamethoxazole (SUT), and tetracycline (TET). *E. coli* ATCC 25922 was used as a quality control strain. Strains were classified as ‘non-susceptible’ when they were resistant or showed intermediate resistance to at least one antimicrobial.

Multidrug resistance (MDR) was defined as the non-susceptibility to at least one agent in three or more antimicrobial classes according to the definition proposed for other bacterial groups [32]. Multiple antimicrobial resistance (MAR) index was calculated using the MAR formula = *a/b*, in which *a* represents the number of antimicrobials to which the tested isolate was non-susceptible and *b* represents the total number of antimicrobials to which the tested isolate was exposed. MAR values higher than 0.2 indicate high-risk sources of antimicrobial contamination in the environment, suggesting a hotspot for antimicrobial resistance [33].

### 2.5. Phenotypic Detection of Extended-Spectrum Beta-Lactamase Producers

*Vibrio* spp. resistant to at least two beta-lactam drugs were tested for the ability to produce extended-spectrum beta-lactamases (ESBL) as previously described [34]. The tested strains were considered positive when it was possible to observe the enhancement of the inhibition zone between the beta-lactams’ disks and the disk containing beta-lactamase inhibitor (amoxicillin-clavulanic acid). *E. coli* ATCC 25922 was used as a negative control and *Klebsiella pneumoniae* ATCC 700603 was used as a positive control for the production of ESBL.

### 2.6. Detection of Beta-Lactam and Heavy Metal Resistance Genes, and Virulence Genes 

Bacterial genomic DNA was obtained using Chelex 100 resin as previously described [35]. ESBL-positive strains were further investigated for the presence of beta-lactam and heavy metal resistance genes. PCR reactions were performed for genes encoding resistance to beta-lactams (*bla*_CTX-M-8_, *bla*_CTX-M-1,2_, *bla*_CTX-M-14_, *bla*_GES_, *bla*_TEM_, and *bla*_SHV_) and were carried out in a total volume of 25 μL using 10 ng of genomic DNA, GO TAQ Green Master Mix (Promega), and 20 pM of each primer (Appendix A). As to the heavy metals, resistance to cadmium, copper, lead, and mercury was investigated by PCR reactions carried out in a total volume of 25 μL using 10 ng of genomic DNA, 12.5 μL of GO TAQ Green Master Mix, 0.2 μL of BSA (50 mg/mL; Sigma-Aldrich, St. Louis, MI, USA), 1.2 μL of Igepal (1.0%; Sigma), and 20 pM of each primer (Appendix A). The presence of such heavy metals is well-established in marine environments and have also been reported in the GB [36,37].

Amplification of the *tdh*, *trh*, and *toxR* genes was performed for strains identified as *V. parahaemolyticus* via MALDI-TOF MS and classified as non-susceptible to antimicrobials. Reactions were carried out in a total volume of 25 μL using 10 ng of genomic DNA, 12.5 μL of GO TAQ Green Master Mix, 0.2 μL of BSA (50 mg/mL), 1.2 μL of Igepal (1.0%), and 20 pM of each primer (Appendix A). Primer sequences and amplification conditions are further detailed in Appendix A.

### 2.7. Data Analysis and Statistics 

Correlations between environmental parameters and the abundance of *Vibrio* spp. (CFU/mL) in each sampling site (as well as from subsurface as from bottom water samples) were calculated by Pearson’s correlations. Additionally, a one-way analysis of variance (ANOVA) was applied to evaluate differences in the abundance of *Vibrio* spp. among the sampling sites and sampling expeditions. All analyses were carried out using the software SPSS version 21.0 (IBM, Armonk, NY, USA), and statistical significance was set to *p* < 0.05.

### 2.8. BOX-PCR

Non-susceptible isolates belonging to the species *V. harveyi* and *V. parahaemolyticus* that remained viable throughout the study were submitted to analysis by BOX-PCR. Apart from being the most frequently isolated species in this study, *V. harveyi* and *V. parahaemolyticus* were selected based on previous literature reports of their pathogenic potential and their resistance characteristics observed in this study. Amplification reactions were performed in a total volume of 25 μL containing 10 ng of genomic DNA, 12.5 μL of GO TAQ Green Master Mix, 0.2 μL of BSA (50 mg/mL), 1.2 μL of Igepal (1.0%), and 1.0 μL of the primer BOXA1R (5′-CTACGGCAAGGCGACGCTGACG-3′) [38]. Amplification conditions consisted of an initial denaturation step for 1 min 30 s at 94 °C; 34 cycles of 30 s at 94 °C, 30 s at 53 °C, and 1 min at 72 °C; final extension for 10 min at 72 °C; and cooling to 4°C. Agarose gel electrophoresis of PCR products was performed using 1.5% agarose in 1 × TBE buffer at 100 V for 3 h at room temperature. Results were organized in matrices indicating the presence or absence of bands (scored as 1 or 0, respectively). Dendrograms were constructed using Dice similarity coefficients and the unweighted pair group method with arithmetic mean (UPGMA) with the software BioNumerics version 7.6 (Applied Maths, Sint-Martens-Latem, Belgium).

## 3. Results

### 3.1. Bacterial Isolation, Characterization, and Identification

In this study, 66 water samples were collected and analyzed, from which 1173 colony forming units (CFU) were isolated on TCBS agar (Figure 1a). Out of these, 14.2% CFU were isolated from site 1 (*n* = 167, being 70 isolated from the subsurface and 97 from the bottom), 15.0% from site 7 (*n* = 175, being 47 isolated from the subsurface and 128 from the bottom), and 70.8% from site 34 (*n* = 831, being 648 isolated from the subsurface and 183 from the bottom). Overall, samples from subsurface waters resulted in a higher isolation percentage (65.2%) when compared to bottom water samples (34.8%) (Figure 1b). From the isolated CFU, 517 colonies were selected according to the characteristics shown on TCBS agar and submitted to further testing.

Viable isolates were submitted to the Gram-staining method and catalase test, revealing 282 Gram-negative bacilli and 348 catalase-positive bacteria. These bacteria were submitted to MALDI-TOF MS identification, from which 192 (68.0%) were assigned to the genus *Vibrio*. Out of these, 39 (20.3%) obtained scores between 2.000 and 2.299, while 153 (79.7%) obtained scores between 1.700 and 1.999. In total, 20 distinct *Vibrio* species were identified, among which *Vibrio harveyi* was the most frequently isolated (44.3%), followed by *Vibrio parahaemolyticus* (16.2%) and *Vibrio mediterranei* (11.4%). In all sampling locations, *V. harveyi* was the most frequently isolated species in subsurface as well as in bottom water samples (Figure 2). Site 7 was characterized by the highest number of distinct *Vibrio* species (*n* = 13) and also the one with the highest rate of *Vibrio* spp. identification (*n* = 72; 37.5%), followed by sites 34 (*n* = 62; 32.3%), and 1 (*n* = 58; 30.2%). At least one potentially pathogenic species was detected in each sampling site and all sampling expeditions. No statistically significant differences were detected between the abundance of *Vibrio* spp. in the sampling sites (*p* = 0.245). Bottom water samples showed the highest *Vibrio* spp. isolation rate (57.8%) when compared to subsurface samples (42.2%).

The highest number of identified *Vibrio* spp. was detected in March, followed by February and April. However, no statistically significant differences were detected between the abundance of *Vibrio* spp. and sampling expeditions (*p* = 0.071). The pathogenic species *V. harveyi* and *V. parahaemolyticus* were detected in all sampling expeditions. Although a selective culture medium for the genus *Vibrio* was used, eight other bacterial genera were identified, being *Photobacterium* spp. and *Aeromonas* spp. the most frequently isolated (Appendix A).

### 3.2. Antimicrobial Susceptibility Profile

From the MALDI-TOF MS identified *Vibrio* spp. isolates, 149 remained viable throughout the study and were thus submitted to the antimicrobial susceptibility tests. The results are summarized in Table 1. Higher non-susceptibility percentages were detected to cefotaxime (25.5%) and amikacin (22.1%). In contrast, the lowest non-susceptibility percentages were detected to imipenem and tetracycline (1.4%). Overall, 63 (42.3%) isolates were classified as non-susceptible to at least one of the tested antimicrobials. MAR indexes ranged from 0.125 to 0.875, with an average of 0.24 and ten strains (6.7%) were classified as MDR. Thirty-one *Vibrio* spp. strains (20.8%) showed MAR values > 0.2. Moreover, four strains (2.7%) were considered ESBL producers, from which two were identified as *V. harveyi* and two as *V. parahaemolyticus*.

As to the sampling sites, site 34 was the one with the highest number of non-susceptible *Vibrio* spp. isolates (*n* = 22; 35.0%), followed by site 1 (*n* = 21; 33.3%) and site 7 (*n* = 20; 31.7%). In addition, site 34 was the one where resistance to all antimicrobial classes was detected. One strain isolated from site 34 identified as *V. parahaemolyticus* and positive for the phenotypical production of ESBL, was resistant to six antimicrobials (amikacin, amoxicillin-clavulanic acid, cefotaxime, ceftazidime, ciprofloxacin, and imipenem) and showed intermediate resistance to one antimicrobial (tetracycline). Thus, this strain was only susceptible to trimethoprim-sulfamethoxazole and its MAR index was 0.875.

### 3.3. Detection of Beta-Lactam Resistance, Heavy Metal Resistance Genes, and Virulence Genes

ESBL-positive strains were further characterized to assess the presence of beta-lactam and heavy metal resistance genes. Genotypes are summarized in Table 2. In our study, strains that harbored beta-lactam resistance genes also carried genes encoding resistance to at least one heavy metal. Among the tested strains, all were characterized by MAR indexes >0.2, and two were multidrug-resistant. Also, all strains were non-susceptible to cefotaxime, a third-generation cephalosporin. Finally, all *V. parahaemolyticus* strains tested for the presence of the *toxR* gene were considered positive (*n* = 11), but none harbored the *tdh* or *trh* genes (*tdh*^−^*trh*^−^).

### 3.4. Physical-Chemical Analysis

Environmental parameters were measured monthly in the three sampling sites in the GB. Their variation, average values, and standard deviation are summarized in Table 3, while all measurements in each sampling site and expedition are available in Appendix A. Results corroborated with the hypothesis that there is a pollution gradient from the inner regions of the GB (marked by higher nutrient concentrations and slower water renewal rates) to its entrance (characterized by higher oceanic influence and lower nutrient concentrations). Site 34 showed not only the highest water temperatures but also the highest concentrations of nitrogen, phosphorus, chlorophyll, pheophytin, and bacterial counts in all sampling expeditions. On the other hand, site 1 was characterized by the lowest concentrations of nitrogen, phosphorus, thermotolerant coliforms, and *E. coli* counts.

Among the analyzed locations, it was observed that the average water temperature was higher in March (27.5 °C) and April (26.9 °C), while salinity levels reached the lowest values in these months (29.3 and 27.4, respectively). Accordingly, higher *Vibrio* spp. isolation rates were observed in March, followed by February and April. The lowest rate of *Vibrio* spp. isolation was in September when water temperature was among the lowest recorded (23.0 °C) and salinity was among the highest (32.4). Both total phosphorus and total nitrogen mean concentrations peaked in October (7.6 and 194.8 µmol/L, respectively), August (7.1 and 148.3 µmol/L), and June (7.0 and 139.8 µmol/L), while the lowest concentrations were observed in November (2.6 and 65.7 µmol/L), September (4.3 and 91.5 µmol/L), and February (4.6 and 91.6 µmol/L). Total bacterial counts peaked in April when the water temperature was relatively high, and salinity was the second-lowest throughout the study period.

### 3.5. Correlations between Environmental Parameters and Vibrio spp. Abundance 

*Vibrio* spp. were present in all sampling sites and sampling expeditions. The highest number of *Vibrio* spp. was identified in March 2019 (*n* = 30), when water temperature was the highest recorded (Figure 3a) and salinity was the second-lowest (Figure 3b). February 2019 was the second month with the highest *Vibrio* spp. count (*n* = 29), followed by April 2019 (*n* = 27). Overall, higher *Vibrio* spp. abundances were observed in the water temperature range from 26.0 to 29.0 °C, salinity range from 30.0 to 35.0, phosphorus concentrations from 1.0 to 5.9 µmol/L, nitrogen concentrations from 0.0 to 20.9 µmol/L, and concentrations of chlorophyll and pheophytin from 0.0 to 5.9 µg/L.

Overall, there was a moderate negative correlation on site 1 (bottom) between *Vibrio* spp. abundance and nitrogen levels (r = −0.698, *p* = 0.017) and a strong negative correlation with water transparency (r = −0.764, *p* = 0.006). On site 7 (subsurface), a moderate negative correlation between *Vibrio* spp. abundance and salinity (r = −0.652, *p* = 0.030), chlorophyll (r = −0.656, *p* = 0.028), thermotolerant coliforms (r = −0.619, *p* = 0.042), and total coliforms (r = −0.695, *p* = 0.018) was observed. Also, site 7 (bottom) displayed a moderate negative correlation among *Vibrio* spp. abundance, thermotolerant coliforms (r = −0.645, *p* = 0.032), and *E. coli* (r = −0.648, *p* = 0.031), while a strong negative correlation was observed for total coliforms (r = −0.765, *p* = 0.006). No statistically significant correlations were identified between the environmental parameters and *Vibrio* spp. abundance on site 34 (Appendix A).

Regarding *Vibrio* isolates classified as non-susceptible to antimicrobials, it was observed that there were moderate positive correlations between these bacteria and chlorophyll (r = 0.621, *p* = 0.041), pheophytin (r = 0.628, *p* = 0.039), and *E. coli* (r = 0.615, *p* = 0.044) on site 7 (subsurface). When site 34 was analyzed, a moderate positive correlation was observed between non-susceptible *Vibrio* spp. and chlorophyll on the bottom water samples (r = 0.666, *p* = 0.025). Interestingly, when we analyzed the non-susceptible *Vibrio* isolates and their correlations with environmental parameters in each sampling expedition, several strong positive and negative correlations were observed, suggesting an association between non-susceptible *Vibrio* spp. and environmental parameters in four months. Here, we highlight that in May 2019, non-susceptible *Vibrio* spp. were significantly correlated with eutrophication indicators. All these results are summarized in Table 4. No statistically significant correlations were detected between non-susceptible *Vibrio* spp. and environmental parameters in the other months. All results obtained in Pearson’s correlation analyses are summarized in Appendix A.

### 3.6. BOX-PCR

Amplification patterns obtained via BOX-PCR were used to estimate the genetic proximity between strains identified as *V. harveyi* (*n* = 21) and *V. parahaemolyticus* (*n* = 11) classified as non-susceptible to antimicrobials and that remained viable throughout the study. Among *V. harveyi* strains, 4 to 12 bands were generated, ranging from 200 to 2000 bp. Results revealed the existence of 16 groups, among which three showed clonal potential. The first group (1) consisted of three strains derived from site 1 (subsurface) but isolated from different sampling expeditions. The second group (2) consisted of three strains isolated from sites 7 (bottom) and 34 (subsurface). These strains were isolated in distinct sampling expeditions. The third group (3) consisted of two strains isolated from the same sampling site and sampling expedition (Figure 4A). Regarding the *V. parahaemolyticus* strains, 1 to 7 bands were generated, ranging from 200 to 3000 bp. Results revealed the existence of nine groups, among which two showed clonal potential. One group (4) consisted of two strains isolated from the same sampling expedition but different sampling sites (7 bottom and 34 subsurface). The second group (5) consisted of strains isolated from the same sampling site and expedition (Figure 4B).

## 4. Discussion

This study provides new insights regarding the occurrence, abundance, and diversity of *Vibrio* spp. isolated from water samples across a pollution gradient in an urban Brazilian estuary. Over one year, *Vibrio* spp. were detected in all sampling sites and expeditions. In total, 20 distinct species were identified, among which 18 have been previously described as potential pathogens. *V. harveyi*, *V. parahaemolyticus*, and *V. mediterranei* were the most frequently identified species. In a previous report, *V. communis, V. parahaemolyticus,* and *V. alginolyticus* were among the most isolated species in GB’s waters according to *pyrH* gene sequence analysis [15]. *V. parahaemolyticus* is a well-known human pathogen, being one of the leading causes of diarrheal diseases associated with seafood consumption worldwide. Importantly, all tested *V. parahaemolyticus* harbored the *toxR* gene, which plays an important role in the regulation of virulence genes and can be associated with virulent strains [39]. The two major virulence factors in this species are *tdh* and *trh*, which encode the thermostable direct hemolysin and the TDH-related hemolysin, respectively. These genes were not detected in our study, which is consistent with previous reports that indicate their low prevalence in environmental strains [40]. However, that does not exclude the pathogenic potential of these strains. Recently, studies have demonstrated clinical *V. parahaemolyticus* isolates lacking *tdh* or *trh* genes, suggesting that other virulence factors are involved in their pathogenesis, while also raising questions as to the use of such genetic markers as predictors of virulence [40,41]. For example, Xu and colleagues (2015) reported that 14% of clinical isolates harbored neither genes, that is, these strains would conventionally be classified as avirulent [42]. Even though hemolysins alone can cause symptoms on the host, other virulence factors may play an important role, such as proteases, siderophores, biofilm-forming ability, and secretion systems [40,42,43,44]. It is also worth mentioning that the pathogenic potential of *Vibrio* spp. might be increased in warmer waters due to the enhanced expression of virulence genes, thus reflecting higher rates of infection and a threat to public health. This is particularly alarming in the framework of climate change, but also in environments characterized by naturally warmer waters, such as the GB, where temperatures above 20 °C were recurrent in all sampling sites during the analyzed period [9,45,46,47].

Reports of *V. harveyi* and *V. mediterranei* in these waters are scarcer. Nonetheless, they are human and animal pathogens, capable of infecting shrimps, oysters, fish, mussels, and many others, leading to a severe negative economic impact on the aquaculture industry [47]. Here we also highlight that, given the paucity of reports of *Vibrio* spp. in the GB, we believe that this may be the first description of the isolation of *V. agarivorans*, *V. brasiliensis*, *V. chagasii*, *V. cyclitrophicus*, *V. fortis*, *V. gigantis*, *V. ichthyoenteri*, *V. navarrensis*, *V. pelagius*, *V. pomeroyi*, *V. rotiferanius*, *V. scophthalmi*, *V. shilonii*, and *V. tasmaniensis* in these waters.

Another important aspect in our findings was the fact that TCBS agar allows the growth of other bacteria, apart from those belonging to the genus *Vibrio*. On account of being a well-known selective media for the isolation of *Vibrio* spp. from clinical and environmental samples, some studies may classify total bacterial counts on TCBS as *Vibrio* spp. [48,49]. However, our results revealed that at least 19.0% of the bacteria isolated on TCBS agar submitted to MALDI-TOF MS identification did not belong to the genus *Vibrio*, being *Photobacterium* and *Aeromonas* the most frequently identified genera. *Photobacterium damselae* belongs to the *Vibrionaceae* family and is also known as an animal and human pathogen [50]. The species of *Aeromonas* identified in our study (*A. caviae, A. hydrophila*, and *A. veronii*) are also considered animal and human pathogens [51]. Other reports have previously described the isolation of non-*Vibrio* bacteria on TCBS agar as well [52,53]. Therefore, the results of phenotypic characterization and identification via MALDI-TOF MS indicate that TCBS agar indeed allows the isolation of other bacterial genera, making it possible to question its so-esteemed selective properties. Hence, it is advisable to use at least one additional identification test to the selective medium.

Since bacteria belonging to the genus *Vibrio* are widely distributed in aquatic environments, human and animal exposure to them is not unusual. For instance, Shaw and colleagues (2015) indicated that swimmers in the Chesapeake Bay (USA) and other people that may enter in contact with those waters experienced significant dermal exposures to *V. vulnificus* and *V. parahaemolyticus* [24]. Considering the pathogenic potential of some species and the fact that antimicrobials are frequently employed to treat infections caused by these microorganisms, it is crucial to better comprehend their antimicrobial susceptibility profiles given the staggering resistance rates worldwide. The World Health Organization (WHO) acknowledges the antimicrobial resistance issue as one of the biggest concerns worldwide, making it harder and more expensive to treat infections, thus resulting in increased mortality rates [54]. Tetracyclines, aminoglycosides, third-generation cephalosporins, and fluoroquinolones are usually employed for the treatment of *Vibrio* spp. infections [55]. In this study, resistance to all the aforementioned drugs was detected, being third-generation cephalosporins and aminoglycosides the classes with the highest resistance percentages. *Vibrio* spp. resistant to such drugs have already been reported in previous studies, not only from water samples but also from fish, shrimp, and oysters, thus indicating a possible dissemination route of drug-resistant bacteria to humans [11]. Furthermore, pathogenic species such as *V. harveyi* and *V. parahaemolyticus* were the ones most frequently associated with multidrug-resistance and ESBL production, especially on sites 34 and 7, where fishing and recreational activities are common. We also highlight the isolation of a strain identified as *V. parahaemolyticus* from site 34 that was multidrug-resistant and positive for the production of ESBL. This strain, along with another isolated from the same sampling site and identified as *V. alginolyticus*, were resistant to imipenem (carbapenem). Carbapenems are considered antimicrobials of last resort since bacteria resistant to this beta-lactam subclass are usually resistant to the majority of other antimicrobials [56]. Previous studies have reported carbapenemase-producing *V. cholerae* in coastal waters in Germany, raising concerns as to the existence of hotspots of antimicrobial resistance in such environments [57,58]. The isolation of these bacteria indicates the risks to human and animal health but other reports on the antimicrobial susceptibility and virulence of *Vibrio* spp. in the GB are utterly scarce.

One of the main factors associated with the emergence and spread of antimicrobial resistance in aquatic environments is sewage. Both treated and untreated sewage have been documented to harbor MDR bacteria, antimicrobial resistance genes (ARGs), and antimicrobial residues from a plethora of classes [59]. However, sustainable sewage destination still represents one of the main challenges for large cities, such as Rio de Janeiro, where it is estimated that sewage from more than 30.0% of the residents in the metropolitan area is not properly collected [60]. Along with sewage discharge, the GB also receives large volumes of industrial effluents, that contribute to the input of heavy metals, hydrocarbons, and nutrients. On account of slower water renewal rates, the inner regions of the GB tend to be characterized by higher concentrations of pollutants. Hence, site 34, the most internal location, is also the most ecologically impacted one. Also, our results showed that most of the ESBL-positive bacteria were isolated from this location, all of which carried beta-lactam and heavy metal resistance genes, as well as high MAR indexes, suggesting an antimicrobial resistance hotspot. Locations submitted to intense anthropogenic pollution provide an environment with strong selective pressure. It has been shown that the presence of antimicrobials and heavy metals, even in sub-inhibitory concentrations, may trigger the emergence and dissemination of antimicrobial resistance, especially because it induces the SOS response in bacteria, favoring mutation and HGT events [61]. Hence, bacteria able to better adapt to such conditions tend to persist and disseminate antimicrobial resistance in the environment. Under this perspective, results obtained via BOX-PCR suggest that clonal groups formed by antimicrobial-resistant *V. harveyi* and *V. parahaemolyticus* can be detected in these waters in different sites and months, thus indicating their persistence in the environment and the relevance of their monitoring.

The detection of heavy metal-resistant bacteria may reflect the levels of contamination in the environment and is often associated with the co-selection of antimicrobial resistance [62]. Regarding the genus *Vibrio*, investigations on the detection of resistance genes are usually described for bacteria isolated from seafood and aquaculture systems. Our study revealed that ESBL-positive strains isolated from water samples carried *bla*_CTX-M_, *bla*_GES_, *bla*_SHV_, and *bla*_TEM_ as well as genes related to copper and mercury resistance. The presence of heavy metals, such as lead, copper, chromium, and zinc has already been described in the GB, especially the northwest region, an area that can be classified as a hotspot of trace metal contamination [63]. Although there are reports of *bla*_CTX-M_, *bla*_SHV_, and *bla*_TEM_ genes in this bacterial genus, there are few reports on *Vibrio* carrying *bla*_GES_ [64,65]. These beta-lactam resistance genes are often detected in *Enterobacteriaceae* and are of clinical relevance. This leads to concerns as to the impact of sewage disposal in aquatic environments, which may provide a source of antimicrobial-resistant microorganisms and reinforce selective pressure, thus contributing to the emergence and dissemination of antimicrobial resistance [66]. The detection of heavy metal resistance genes in antimicrobial-resistant strains raises attention to the possibility of the presence of mobile genetic elements and integrons that could mediate HGT events and disseminate resistance. Moreover, our findings indicate that bacteria derived from sewage, such as coliforms, co-exist with *Vibrio* spp., especially on site 34, and they could play a pivotal role in the HGT of ARGs to the indigenous bacterial community, notably *Vibrio* spp., a bacterial genus known for its ‘genomic promiscuity’ [2].

Our results further support the hypothesis that the study of *Vibrio* spp. is an interesting alternative to the monitoring of pathogenic species and microbial resistance in aquatic environments. Hence, another important aspect would be to consider environmental parameters and how they correlate with the bacterial community. The alignment of microbiological and environmental data provides helpful information that could be used to estimate the risks to human health and predict outbreaks [67,68]. Over one year, 12 environmental parameters were analyzed in three sites in Guanabara Bay. Higher *Vibrio* spp. isolation rates occurred in months characterized by higher water temperatures and lower salinities. Although correlations between *Vibrio* spp. and water temperature were not detected, negative correlations with salinity were established, which is consistent with previous reports in Sidney (Australia), and also in the Guanabara Bay [69,70]. High concentrations of chlorophyll, total phosphorus, and low water transparency, as observed in the most impacted site in our study, are indicators of eutrophication and cyanobacterial and phytoplankton proliferation, suggesting higher probabilities of adverse health effects [71]. These blooms may be associated with the growth of potentially pathogenic bacteria, including *Vibrio* spp., as observed by Asplund and colleagues (2011), raising concerns as to the safety of these recreational waters [72].

Currently, microbiological monitoring of water quality indicators is mainly achieved by the detection of *Enterococcus*, coliforms, and *E. coli*, in Brazil [73] as well in other countries that rely on the Bathing Water Directive (2006/7/EC) [74]. However, this strategy overlooks other threats to human health, such as the presence of antimicrobial-resistant microorganisms, other potential pathogens, and virulence markers. In view of such limitations, the WHO has proposed several additional recommendations regarding the evaluation of water quality parameters, among which the genus *Vibrio* has been included as a matter of concern, especially on account of emerging infections related to the increase in water temperatures [68,71]. Importantly, as seen in our study, *Vibrio* spp. may not always correlate with the occurrence of conventional fecal pollution indicators [75]. Although the present study focused on *Vibrio* spp., our results may well have a bearing on other pathogenic bacteria found in aquatic environments. Taken together, these findings support strong recommendations to the monitoring of other bacterial pathogens in aquatic environments and extending their characterization to also encompass the search of antimicrobial resistance traits.

The presence of *Vibrio* spp. non-susceptible to antimicrobials is also alarming in the context of plastic pollution. Previous studies in the GB have already reported the presence of expressive amounts of plastics and microplastics throughout all of its extensions, including sites near the ones analyzed in our study [16,76]. Plastics are emerging pollutants that provide relatively stable habitats for microorganisms, which in turn are able to form biofilms on such surfaces. *Vibrio* spp. have been confirmed to colonize microplastics, where they can be found in high abundance [3]. Recently, Kesy and colleagues (2021) demonstrated that *Vibrio* spp. are colonizers of microplastics in the Baltic Sea, an environment that is also greatly impacted by anthropogenic pollution. In this same study, the authors also observed that the abundance of *Vibrio* on such particles was greater in areas closer to major cities [77]. Therefore, we hypothesize that *Vibrio* spp. could be receiving ARGs from bacteria discharged via sewage in the GB and being further transported with the tide via plastic debris or in their planktonic form throughout the Bay, thus posing a serious risk to the health of citizens that are exposed to these waters. These plastic particles colonized by bacteria are a source of contamination and could also be ingested by marine animals in the GB, being a risk to local fauna and to the health of those who may consume seafood from this region.

Notwithstanding the analysis of limited water samples using a selective medium and the fact that our observations relied on culture-dependent methods, several clinically relevant traits were detected in a key aquatic bacterial genus throughout the whole study period. Although culture-independent methods such as real-time polymerase chain reactions may be faster when compared to culture-dependent approaches, they may not reflect the viable bacterial community in a given ecosystem and may lead to an overestimation of health risks. This is particularly relevant when it comes to the genus *Vibrio* since several of its members are known to exist in a viable but non-culturable state [68]. This approach was therefore useful to suggest that these characteristics are present in GB’s waters and may reflect the bacterial adaptive process in an ecosystem severely impacted by anthropogenic pollution.

## 5. Conclusions

To the best of our knowledge, this is the first comprehensive assessment of the antimicrobial susceptibility of *Vibrio* spp. isolated from GB’s waters. The detection of MDR bacteria, ESBL-producers, and bacteria harboring beta-lactam and heavy metal resistance genes, along with the presence of potential pathogens, high levels of eutrophication, and fecal contamination indicators, suggests risks to public health. Given the dearth of continuous monitoring of water habitats for the presence of *Vibrio* spp., especially non-*cholerae Vibrio* species, we reinforce that risk-assessment research is of paramount importance. By doing so, exposure to aquatic environments and consumption of seafood can be limited to minimize the risks of infection. Thereupon, our findings shed new light on the matter of risk-assessment research in the GB and support the need for urgent public measures aimed at more efficient sanitation services, the decrease in pollution levels, and the promotion of antimicrobial stewardship, not only in Brazil but also in other countries around the world.

## Figures and Tables

**Figure 1 microorganisms-09-01007-f001:**
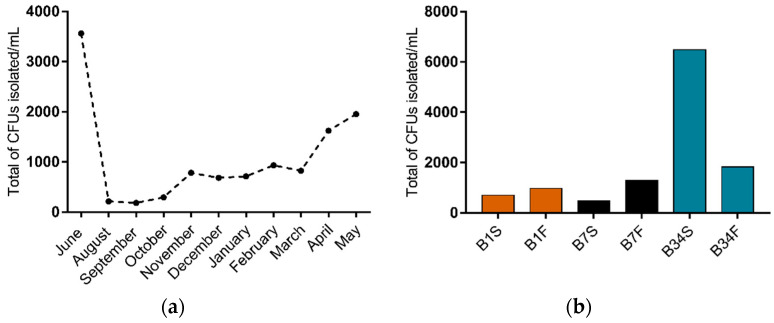
Distribution of CFU isolated on TCBS agar from Guanabara Bay’s waters over the course of one year. (**a**) Distribution of total CFU/mL isolated in each sampling expedition from June/2018 to May/2019. (**b**) CFU/mL isolated in each sampling site, from the subsurface as well as from the bottom water samples. B1S: site 1 subsurface; B1F: site 1 bottom; B7S: site 7 subsurface; B7F: site 7 bottom; B34S: site 34 subsurface; B34F: site 34 bottom.

**Figure 2 microorganisms-09-01007-f002:**
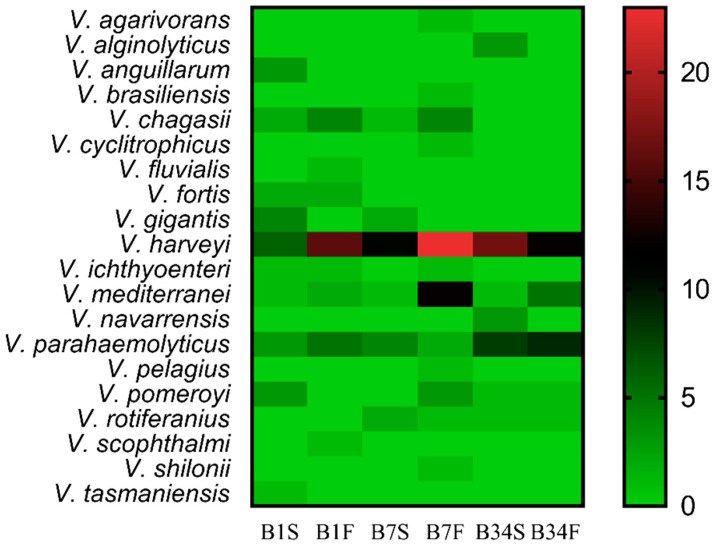
Distribution of *Vibrio* spp. among the sampling sites. Heatmap of the distribution of *Vibrio* species identified by MALDI-TOF MS in each sampling location, from the subsurface as well as from the bottom water samples. The color gradient key displays a linear scale of the MALDI-TOF MS identified *Vibrio* coverage as a measure of the absolute abundance in each sampling site. B1S: site 1 subsurface; B1F: site 1 bottom; B7S: site 7 subsurface; B7F: site 7 bottom; B34S: site 34 subsurface; B34F: site 34 bottom.

**Figure 3 microorganisms-09-01007-f003:**
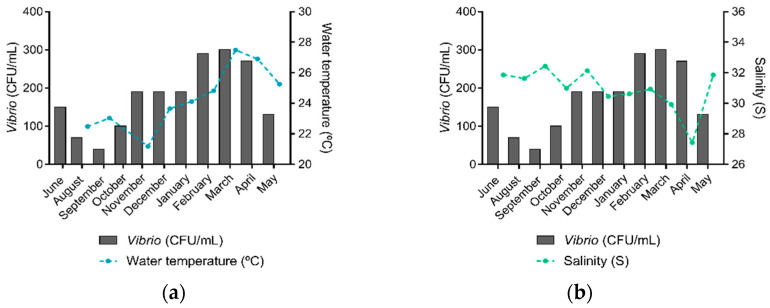
Water temperature, salinity, and *Vibrio* spp. abundance in the Guanabara Bay in each sampling expedition over the course of one year. (**a**) Average water temperature (°C) and the total of *Vibrio* spp. (CFU/mL) identified and (**b**) Average salinity (S) levels and the total of *Vibrio* spp. (CFU/mL) identified in each sampling expedition.

**Figure 4 microorganisms-09-01007-f004:**
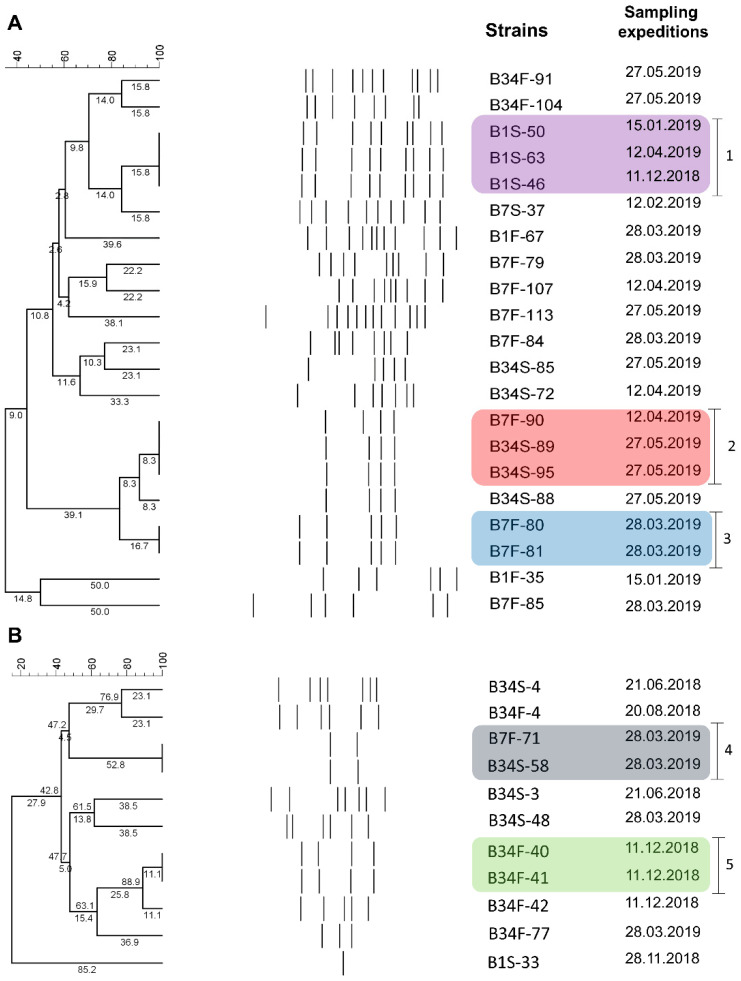
BOX-PCR results. Cluster analysis of banding patterns of *V. harveyi* (**A**) and *V. parahaemolyticus* (**B**) strains non-susceptible to antimicrobials and their respective isolation dates. B1S: site 1 subsurface; B1F: site 1 bottom; B7S: site 7 subsurface; B7F: site 7 bottom; B34S: site 34 subsurface; B34F: site 34 bottom.

**Table 1 microorganisms-09-01007-t001:** Antimicrobial susceptibility of *Vibrio* spp. isolated over a one-year period.

Antimicrobial	Number of Isolates		
	S ^1^	I ^2^	R ^3^
Amikacin	116 (78.0%)	12 (8.0%)	21 (14.0%)
Cefotaxime	111 (74.5%)	11 (7.4%)	27 (18.1%)
Imipenem	147 (98.6%)	0	2 (1.4%)
Amoxicillin-clavulanic acid	133 (89.3%)	7 (4.7%)	9 (6.0%)
Ciprofloxacin	135 (90.6%)	11 (7.4%)	3 (2.0%)
Ceftazidime	134 (89.9%)	4 (2.7%)	11 (7.4%)
Tetracycline	147 (98.6%)	1 (0.7%)	1 (0.7%)
Trimethoprim-sulfamethoxazole	146 (98.0%)	0	3 (2.0%)

^1^ S–susceptible; ^2^ I–intermediate; ^3^ R–resistant.

**Table 2 microorganisms-09-01007-t002:** ESBL-positive *Vibrio* spp., their respective antimicrobial non-susceptibility, resistance genes detected, and MAR index.

Strains	Antimicrobial Non-Susceptibility	Resistance Genes	MAR Index
*V. harveyi* B7F-82 *	AMI, CAZ, CIP, CTX	*bla*_TEM_, *bla*_SHV_, *merA*, *A1F*/*A1R*	0.500
*V. parahaemolyticus* B34S-4 ^1^	CAZ, CTX	*bla*_GES_, *bla*_CTX-M-1,2_, *copA*, *cusB*	0.250
*V. parahaemolyticus* B34F-4 ^1^ *	AMC, AMI, CAZ, CIP, CTX, IPM, TET	*bla*_SHV_, *merA*, *A1F*/*A1R*	0.875
*V. harveyi* B34F-73	AMC, AMI, CTX	*bla*_TEM_, *bla*_CTX-M-1,2_, *copA*	0.375

* MDR strains; ^1^
*toxR*-positive strains.

**Table 3 microorganisms-09-01007-t003:** Environmental parameters from 2018 to 2019 in Guanabara Bay’s waters. Minimum, maximum, average values and standard deviation of environmental parameters in each sampling site.

Environmental Parameters	Sampling Sites in the GB *		
	Site 1	Site 7	Site 34
Water temperature (°C)	16.80–27.22 (22.80 ± 3.09)	17.30–28.60 (23.61 ± 3.30)	22.60–31.00 (26.47 ± 2.59)
Air Temperature (°C)	21.70–31.80 (26.70 ± 2.70)	21.00–30.0 (26.45 ± 2.83)	24.60–33.50 (28.95 ± 2.71)
Water transparency (m)	1.40–5.80 (2.94 ± 1.33)	0.60–5.80 (1.79 ± 1.23)	0.40–1.05 (0.73 ± 0.19)
Salinity (S)	30.08–35.73 (33.8 ± 1.39)	26.28–35.10 (32.7 ± 2.21)	12.61–31.68 (26.12 ± 5.30)
Total phosphorus (µmol/L)	0.75–2.74 (1.43 ± 0.57)	0.89–7.92 (2.34 ± 1.82)	4.59–28.17 (13.01 ± 7.07)
Total nitrogen (µmol/L)	8.84–44.53 (24.61 ± 9.69)	9.73–150.35 (44.32 ± 37.7)	90.63–588.66 (280.85 ± 156.39)
Chlorophyll (µg/L)	1.07–28.07 (8.70 ± 7.67)	0.94–193.13 (22.60 ± 43.27)	7.02–310.07 (79.52 ± 87.20)
Pheophytin (µg/L)	1.46–7.17 (3.17 ± 1.59)	2.28–19.25 (5.46 ± 4.02)	0.94–48.18 (15.81 ± 12.33)
Thermotolerant coliforms (MPN/100 mL)	2–700 (109.95 ± 172.38)	2– 1600 (137.03 ± 365.89)	18–920,000 (70,432.18 ± 198,491.08)
Total coliforms (MPN/100 mL)	2–9200 (539.31 ± 1941.74)	2–1600 (163.42 ± 371.05)	18–920,000 (89,111.27 ± 207,852.28)
*E. coli* (MPN/100 mL)	13–330 (68.36 ± 87.98)	2–920 (99.76 ± 246.47)	18–920,000 (58,911.72 ± 196,373.66)
Heterotrophic bacteria (CFU/mL)	10–244,500 (15,532.63 ± 54,988.25)	18–82,500 (4142.5 ± 17,526.08)	465–1,360,000 (143,427.5 ± 331,527.87)

* Guanabara Bay’s water samples, from the subsurface as well as from the bottom, in each sampling site, obtained from June 2018 to May 2019.

**Table 4 microorganisms-09-01007-t004:** Pearson’s correlations analyses between environmental parameters and antimicrobial non-susceptible *Vibrio* spp. abundances (CFU/mL) in four months.

Environmental Parameters	Antimicrobial Non-Susceptible *Vibrio* spp.							
	August 2018		January 2019		February 2019		May 2019	
	r	*p*	r	*p*	r	*p*	r	*p*
Air temperature	0.837 *	0.038	−0.542	0.266	−0.612	0.196	0.820 *	0.046
Water temperature	0.789	0.062	−0.840 *	0.036	−0.693	0.127	0.956 **	0.003
Salinity	0.867 *	0.025	0.663	0.151	0.880 *	0.021	−0.878 *	0.021
Total phosphorus	0.865 *	0.026	−0.517	0.293	−0.830 *	0.041	0.961 **	0.002
Total nitrogen	0.862 *	0.027	−0.507	0.305	−0.814 *	0.049	0.955 **	0.003
Water transparency	−0.459	0.359	−0.353	0.493	0.698	0.123	−0.558	0.250
Chlorophyll	0.866 *	0.026	−0.512	0.299	−0.852 *	0.031	0.817 *	0.047
Pheophytin	0.263	0.615	−0.583	0.225	−0.791	0.061	0.152	0.774
Thermotolerant coliforms	0.804	0.054	−0.533	0.276	−0.775	0.070	0.936 **	0.006
Total coliforms	0.804	0.054	−0.533	0.276	−0.775	0.070	0.948 **	0.004
*E. coli*	0.807	0.052	−0.533	0.276	−0.775	0.070	0.946 **	0.004
Heterotrophic bacteria	0.801	0.055	−0.449	0.371	−0.790	0.061	0.944 **	0.005

r indicates the Pearson’s correlation coefficient and *p* indicates the *p*-value. * indicates a significant correlation at the 0.05 level; ** indicates a significant correlation at the 0.01 level.

## Data Availability

Not applicable.

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
