# Peer review of "Vibrio Species in an Urban Tropical Estuary: Antimicrobial Susceptibility, Interaction with Environmental Parameters, and Possible Public Health Outcomes"

_microorganisms, 2021, doi:10.3390/microorganisms9051007_

Round 1

Reviewer 1 Report

The paper represents an interesting study worthy to be published about the antibiotic susceptibility of some cultivable Vibrio species isolated around an year from a heavy polluted marine area subjected to recreational use; the research appears, also, able to relate the obtained results with some pathogenic potentialities of the isolated bacteria and a large amount of environment parameters as well from a public health perspective. Only minor concerns have been discovered.

Line 84: the term freshwater is not appropriate to indicate Atlantic Ocean waters.

Line 112: The depth of the different places subjected to samplings is not indicated

Line 128: a bibliographic reference of the microbiological determinations is lacking

Author Response

Author's Response to Reviewer’s Comments

Reviewer: 1

The paper represents an interesting study worthy to be published about the antibiotic susceptibility of some cultivable Vibrio species isolated around an year from a heavy polluted marine area subjected to recreational use; the research appears, also, able to relate the obtained results with some pathogenic potentialities of the isolated bacteria and a large amount of environment parameters as well from a public health perspective. Only minor concerns have been discovered.

Line 84: the term freshwater is not appropriate to indicate Atlantic Ocean waters.

Line 112: The depth of the different places subjected to samplings is not indicated

Line 128: a bibliographic reference of the microbiological determinations is lacking

Author’s reply: Dear reviewer, thank you for your review. We have changed “freshwater” to “water” (page 2, line 83).

As to the depth of the sampling, they are described on page 3, line 111: “Subsurface (1-2 m depth) and bottom waters (6-20 m depth) were sampled...”

Regarding the bibliographic reference to the microbiological determinations, we agree and have included the following reference: Baird, R.B., Standard methods for the examination of water and wastewater, 23rd ed; American Public Health Association, American Water Works Association, Water Environment Federation, 2017 (please, see page 2, line 131).

Should any question arise, please do not hesitate to contact me. Thank you very much for your consideration.

Best regards,

Reviewer 2 Report

  1. Abstract: Over one year, water samples from three locations along 17 a pollution gradient were investigated. Isolates were identified by MALDI-TOF mass spectrometry, 18 revealing 20 Vibrio species, including several potential pathogens. I suggest mentioning the total number of water samples from where 20 Vibrio species were isolated or identified.
  2. All sampling expeditions took place in the morning period and on days with little or no tidal influence. At each sampling site, water samples from the subsurface and the bottom were collected in appropriate sterile plastic bottles and immediately transported to the laboratory in an icebox. In the methodology section number of samples collected from surface and bottom water is scared. How much water samples were collected and how and using what kind of apparatus from the subsurface and bottom water is not clear.
  3. The score values obtained were interpreted according to the manufacturer’s guidelines: ≥ 2.300 indicated reliable identification at species level; 2.299 – 2.000 reliable identification at genus level and likely identification for species; 1.700 – 1.999 reliable identification solely at genus level; and < 1.699 was not considered reliable for identification. E. coli DH5α was used as a quality control strain. The range (2.299 – 2.000 ) is not in correct order.
  4. coli ATCC 25922 and Klebsiella pneumoniae ATCC 700603 were used as negative and positive controls, respectively. Which one used as positive and which one used as negative it should be clear if used both simultaneously then also should need to clear the use of two different controls.
  5. As to the heavy metals, resistance 192 to cadmium, copper, lead, and mercury was investigated by PCR reactions carried out in 193 a total volume of 25 μL using 10 ng of genomic DNA, 12.5 μL of GO TAQ Green Master 194 Mix, 0.2 μL of BSA (50 mg/mL; Sigma), 1.2 μL of Igepal (1.0%; Sigma), and 20 pM of each 195 primer (Table S2). There are other heavy metals as well then why only resistance against cadmium, copper, lead, and mercury were focused on? I suggest it should be clear by giving possible reasons according to your study purpose or area.
  6. Bottom water samples showed the highest Vibrio spp. isolation rate (57.8%) when compared to subsurface samples (42.2%). Two sampling sites showed a higher abundance of vibrio spp in subsurface water and only one site has the highest vibrio spp. So instead of giving a general conclusion that bottom water samples highest vibrio spp compared to surface samples. I suggest it will be better to give the distribution of vibrio spp in the bottom and subsurface based on sampling sites.
  7. Figure 4. clustering of strains and sampling expeditions missed the positive control used in this study. I suggest including the controls in the clustering approach to make sure how the isolated species are similar or dissimilar from the controls.

Author Response

Author's Response to Reviewer’s Comments

Reviewer: 2

Author’s reply: The authors appreciate all the concerns raised by the reviewer, his excellent and throughout assessment of our manuscript. We replied every comment and correction below, discussing whenever possible.

Abstract: Over one year, water samples from three locations along a pollution gradient were investigated. Isolates were identified by MALDI-TOF mass spectrometry, revealing 20 Vibrio species, including several potential pathogens. I suggest mentioning the total number of water samples from where 20 Vibrio species were isolated or identified.

Author’s reply: The total number of water samples has been incorporated in the abstract (page 1, line 16). We have also changed “concentrations of phosphorus and nitrogen” to “phosphorus and nitrogen concentrations” (page 1, line 24) and “may” to “might” (page 1, line 27) for better readability.

All sampling expeditions took place in the morning period and on days with little or no tidal influence. At each sampling site, water samples from the subsurface and the bottom were collected in appropriate sterile plastic bottles and immediately transported to the laboratory in an icebox. In the methodology section number of samples collected from surface and bottom water is scared. How much water samples were collected and how and using what kind of apparatus from the subsurface and bottom water is not clear.

Author’s reply: We have included the number of water samples collected on page 3, lines 112 and 113. “…66 water samples (33 from the subsurface and 33 from the bottom)...”

The description of water sampling is present on page 3, lines 122 to 125. We have also included the apparatus used in the sampling expeditions on line 124, which was used both for the sampling of subsurface and bottom water samples. The water samples were collected using a 5 L Niskin bottle (General Oceanics, Miami).

The score values obtained were interpreted according to the manufacturer’s guidelines: ≥ 2.300 indicated reliable identification at species level; 2.299 – 2.000 reliable identification at genus level and likely identification for species; 1.700 – 1.999 reliable identification solely at genus level; and < 1.699 was not considered reliable for identification. E. coli DH5α was used as a quality control strain. The range (2.299 – 2.000) is not in correct order.

Author’s reply: We thank the reviewer for pointing this out. We have revised and changed “2.299 – 2.000” to “2.000 to 2.299” (page 4, lines 156 and 157).

  1. coli ATCC 25922 and Klebsiella pneumoniae ATCC 700603 were used as negative and positive controls, respectively. Which one used as positive and which one used as negative it should be clear if used both simultaneously then also should need to clear the use of two different controls.

Author’s reply: We have changed “E. coli ATCC 25922 and Klebsiella pneumoniae ATCC 700603 were used as negative and positive controls, respectively” to “E. coli ATCC 25922 was used as a negative control and Klebsiella pneumoniae ATCC 700603 was used as a positive control for the production of ESBL.” (page 4, lines 185-187)

As to the heavy metals, resistance to cadmium, copper, lead, and mercury was investigated by PCR reactions carried out in a total volume of 25 μL using 10 ng of genomic DNA, 12.5 μL of GO TAQ Green Master Mix, 0.2 μL of BSA (50 mg/mL; Sigma), 1.2 μL of Igepal (1.0%; Sigma), and 20 pM of each primer (Table S2). There are other heavy metals as well then why only resistance against cadmium, copper, lead, and mercury were focused on? I suggest it should be clear by giving possible reasons according to your study purpose or area.

Author’s reply: We have included a brief explanation on page 5, lines 198 and 199 regarding the presence of the investigated heavy metals in marine environments in general. We have also included two references that have reported the presence of such metals in the studied area.

Bottom water samples showed the highest Vibrio spp. isolation rate (57.8%) when compared to subsurface samples (42.2%). Two sampling sites showed a higher abundance of Vibrio spp. in subsurface water and only one site has the highest Vibrio spp. So instead of giving a general conclusion that bottom water samples highest Vibrio spp. compared to surface samples. I suggest it will be better to give the distribution of Vibrio spp. in the bottom and subsurface based on sampling sites.

Author’s reply: We have included the distribution of Vibrio spp. isolated in each sampling site both from subsurface and bottom water samples on page 5 (lines 234 to 237) as follows: “In this study, 66 water samples were collected and analyzed, from which 1,173 colony forming units (CFU) were isolated on TCBS agar (Figure 1a). Out of these, 14.2% CFU were isolated from site 1 (n = 167, being 70 isolated from the subsurface and 97 from the bottom), 15.0% from site 7 (n = 175, being 47 isolated from the subsurface and 128 from the bottom), and 70.8% from site 34 (n = 831, being 648 isolated from the subsurface and 183 from the bottom)”.

Figure 4. clustering of strains and sampling expeditions missed the positive control used in this study. I suggest including the controls in the clustering approach to make sure how the isolated species are similar or dissimilar from the controls.

Author’s reply: DNA fingerprint patterns obtained by BOX-PCR (figure 4) were used to estimate the relative degrees of similarity among isolates identified as V. harveyi and V. parahaemolyticus and to help determine whether isolates from distinct sampling sites and expeditions were clonally related. As described by Versalovic et al., 19941, by performing this technique several amplicons of different sizes are fractionated by electrophoresis and enable the establishment of DNA fingerprint patterns specific for individual bacterial strains. DNA fingerprint patterns can then be compared using the Dice similarity coefficient and the unweighted pair group method with arithmetic mean (UPGMA) to estimate relative degrees of similarity among isolates. Hence, the analysis of isolates by BOX-PCR does not require control strains, since our main objective is to compare the isolated strains between themselves, determine their genetic proximity and formulate hypotheses as to their persistence in the environment throughout time. Negative controls were maintained in the amplification reactions.

1Versalovic, J., Schneider, M., De Bruijn, F.J., Lupski, J. R. (1994). Genomic fingerprinting of bacteria using repetitive sequence-based polymerase chain reaction. Methods in Molecular and Cellular Biology, 5(1), 25-40.

Should any question arise, please do not hesitate to contact me. Thank you very much for your consideration.

Best regards,

Marinella Silva Laport, on behalf of all authors.
